# Volatile organic compounds in headspace characterize isolated bacterial strains independent of growth medium or antibiotic sensitivity

**Kim F. H. Hintzen**[1,2,3], **Lionel Blanchet**[1,3], **Agnieszka Smolinska**[1,3], **Marie-Louise Boumans**[3,4], **Ellen E. Stobberingh**[3,4], **Jan W. Dallinga**[1,3], **Tim Lubbers**[2,5], **Frederik-Jan van Schooten**[1,3], **Agnes W. Boots**[1,3]*

1 Department of Pharmacology and Toxicology, Maastricht University, Maastricht, The Netherlands, 2 Department of Surgery, Maastricht University Medical Centre, Maastricht, The Netherlands, 3 NUTRIM School of Nutrition and Translational Research in Metabolism, Maastricht University, Maastricht, The Netherlands, 4 Department of Medical Microbiology, Maastricht University Medical Centre, Maastricht, The Netherlands, 5 GROW School for Oncology and Developmental Biology, Maastricht University, Maastricht, The Netherlands

* a.boots@maastrichtuniversity.nl

**Data Availability Statement:** All relevant data are within the manuscript and its Supporting Information files.

## Abstract

### Introduction

Early and reliable determination of bacterial strain specificity and antibiotic resistance is critical to improve sepsis treatment. Previous research demonstrated the potential of headspace analysis of volatile organic compounds (VOCs) to differentiate between various microorganisms associated with pulmonary infections *in vitro*. This study evaluates whether VOC analysis can also discriminate antibiotic sensitive from resistant bacterial strains when cultured on varying growth media.

### Methods

Both antibiotic-sensitive and -resistant strains of *Pseudomonas aeruginosa*, *Staphylococcus aureus* and *Klebsiella pneumonia* were cultured on 4 different growth media, i.e. Brain Heart Infusion, Marine Broth, Müller-Hinton and Trypticase Soy Agar. After overnight incubation at 37˚C, the headspace air of the cultures was collected on stainless steel desorption tubes and analyzed by gas chromatography *time-of-flight* mass spectrometry (GC-*tof*-MS). Statistical analysis was performed using regularized multivariate analysis of variance and cross validation.

### Results

The three bacterial species could be correctly recognized based on the differential presence of 14 VOCs (p<0.001). This discrimination was not influenced by the different growth media. Interestingly, a clear discrimination could be made between the antibiotic-resistant and -sensitive variant of *Pseudomonas aeruginosa* (p<0.001) based on their species-specific VOC signature.

**Funding:** T. Lubbers (MLDS) Maag Lever Darm Stichting career development grant CDG16-12 supported by Dutch Digestive Foundation (https://www.mlds.nl/) The funders had no role in study design, data collection and analysis, decision to publish, or preparation of the manuscript.

## Conclusion

This study demonstrates that isolated microorganisms, including antibiotic-sensitive and -resistant strains of *Pseudomonas aeruginosa*, could be identified based on their excreted VOCs independent of the applied growth media. These findings suggest that the discriminating volatiles are associated with the microorganisms themselves rather than with their growth medium. This study exemplifies the potential of VOC analysis as diagnostic tool in medical microbiology. However, validation of our results in appropriate *in vivo* models is critical to improve translation of breath analysis to clinical applications.

## Introduction

Sepsis is a major health problem worldwide and a severe condition that can lead to septic shock, multiple organ failure and even death [1]. It is defined as a life-threatening organ dysfunction caused by a dysregulated host response to infection of predominantly the respiratory tract, urinary tract or abdomen [2, 3]. Elderly patients with comorbidities, including chronic obstructive pulmonary disease (COPD) and coronary heart diseases, are particularly vulnerable to develop pneumonia upon encountering a respiratory tract infection [4, 5]. Inadequate identification of the causative pathogen, and subsequent inadequate treatment, increases the chance to develop bacteremia and sepsis. The most common causes of pneumonia are respiratory microorganisms including the bacteria *Pseudomonas aeruginosa*, *Staphylococcus aureus* and *Klebsiella pneumonia* [4–6]. For all three forms of pneumonia, being community-acquired pneumonia (CAP), nosocomial pneumonia (NP) or ventilator-associated pneumonia (VAP), bacteremia is associated with prolonged admission at the intensive care unit (ICU) and increased mortality [5]. In the majority of septic patients, the exact causative pathogen is unknown, leading to treatment with broad-spectrum antibiotics which is too unspecific. Both inadequate and excessive antibiotic treatment with broad-spectrum antibiotics are associated with a higher mortality [2, 7]. Additionally, antimicrobial resistance (AMR) is a growing burden in health care [8], warranting the possible misuse of antibiotics to be kept to a minimum. Therefore, early and reliable determination of strain specificity and antibiotic resistance is critical to improve early adequate treatment and is essential to improve patient outcome.

Bacteria in the human body, both commensal as well as pathogenic bacteria, produce volatile organic compounds (VOCs) as part of their species-specific metabolism [9–12]. Both single VOCs [13, 14] and VOC profiles [9, 15, 16] can be used to distinguish different bacteria. VOCs are mostly carbon based molecules that are excreted in, amongst others, feces, urine and exhaled breath [17], thereby offering an opportunity to trace bacterial infections in the human body. The major advantage of exhaled air analysis is that sampling is non-invasive and has a low impact on patients. Exhaled breath can be collected frequently and analyzing the VOC content has a proven diagnostic power related to various respiratory diseases including COPD, asthma and interstitial lung diseases [18–21]. Measuring alterations in exhaled VOCs could therefore be a promising alternative approach to improve the diagnosis of respiratory infections [22]. Studies from our lab as well as other groups have already shown that patients with VAP can be distinguished from non-VAP patients on the ICU based on a distinct VOC profile that could be related to the causative pathogen [23–25].

Consequently, the aim of the current *in vitro* study is to investigate whether analysis of VOCs can be used to discriminate antibiotic-sensitive and–resistant respiratory pathogens

cultured on varying growth media. The primary hypothesis is that these bacteria induce unique VOC profiles independent of their growth media. The second hypothesis is that the cellular mechanisms of antibiotic resistant bacteria result in an altered VOC profile compared to that of their antibiotic sensitive subtype.

## Material and methods

### Materials and samples

Three different bacterial strains were selected: *Klebsiella pneumonia* (*K. pneumonia*, ATCC 700683), *Pseudomonas aeruginosa* (*P. aeruginosa*, ATCC 27853 and a clinical isolate) and *Staphylococcus aureus* (*S. aureus*, clinical isolate). For all strains, both an antibiotic-sensitive and an antibiotic-resistant strain were included: an extended spectrum β-lactamase (ESBL)-resistant and an ESBL-sensitive for both *P. aeruginosa* and *K. pneumonia* and a methicillin-resistant (MRSA, clinical isolate) and a methicillin-sensitive (MSSA, clinical isolate) strain for *S. aureus*. Antibiotic resistance was tested phenotypically by determining the minimum inhibitory concentration (MIC) of the specific antibiotics, e.g. methicillin and oxacillin for MRSA and MSSA and ESBL for the resistant and sensitive strains of both *K. pneumonia* and *P. aeruginosa*. For MRSA and MSSA, the genotype was additionally tested by analyzing the presence of mec A (the gene responsible for methicillin resistance) as well as by performing multi locus sequence typing (MLST) and *staphylococcus aureus* protein A (Spa typing) to determine whether the two substrains were related. For the two strains of *K. pneumonia*, the genotype was additionally tested by matching for MLST and RAMAN clustering.

All selected substrains were cultured in 4 different growth media, i.e. Brain Heart Infusion (BHI), Marine Broth (MB), Müller-Hinton (MH) and Trypticase Soy Agar (TSA) with 24 repeats per strain per medium. For *P. aeruginosa*, 12 repeats of both the ATCC and clinical isolate were performed. At the time of VOC sampling, the average CFU/ml was $10^9$/ml for each genus as determined by serial dilutions of the growth media. No additional media controls were added to the experimental procedure based on our previous study that already revealed methodological standardization as the medium control group was well separated from the different bacteria [9].

### Sample collection and analysis

All strains were cultured on Columbia III Agar plates with 5% sheep blood (Becton-Dickinson) and incubated overnight at 37˚C. Afterwards, the bacteria were transferred into 4.5 mL sterile Brain Heart Infusion broth (CM0225 Oxoid) and grown for another 4 hours at 37˚C under constant agitation. Next, 0.5 mL of the culture was transferred into 100 mL of one of the 4 selected sterile media (BHI CM0225 Oxoid; MH CM0405 Oxoid; TSA CM0131 Oxoid and MB dd15 Difco) in 1L culture flasks.

After overnight incubation at 37˚C, overhead air (headspace) was sampled during 15 minutes for VOC analysis as described previously [9]. Briefly, by flushing the flasks with high-grade nitrogen the VOCs present in the headspace of the cultures were transported onto stainless two-bed sorption tubes, packed with carbograph 1TC/Carbopack X (Markes International, Llantrisant, Wales, UK), under standardized conditions.

The VOC content of all tubes was determined using gas chromatography *time-of-flight* mass spectrometry (GC-*tof*-MS) as previously published [9, 26]. In short, the desorption tubes were placed inside a thermal desorption unit (Marks Unity desorption unit, Markes International) and heated to 270˚C in order to release all trapped VOCs onto the gas chromatography capillary column (RTX-5ms, 30 m x -.25 mm 5% diphenyl, 95% dimethylsiloxane capillary, film thickness 1.0 μm) followed by separation by the GC (ThermoFisher Scientific, Austin,

Texas, USA) and detection by a *tof*-MS (Thermo Electron Tempus Plus *time-of-flight* mass spectrometer, ThermoFisher Scientific, Austin, Texas, USA). Settings for this separation and detection were similar to those previously published [9, 26].

### Data pre-processing

Before the statistical analysis, the raw GC-*tof*-MS data were preprocessed [27]. Uninformative parts of each chromatogram, (retention time either <1.3 or >23 minutes) were removed. The data quality was improved by denoising (based on wavelets decomposition [28]) and baseline correction. P-splines [29] were used to estimate the shape of the baseline and subtract it from the denoised chromatogram.

A list of chromatographic peaks was established for each chromatogram. These peaks needed to be matched from sample to sample. To this end, the Pearson correlation between the mass spectra, associated to the different chromatographic peaks, were calculated. A high correlation (superior to 85%) indicated that two peaks correspond to the same chemical compound. This information allowed constructing a data table where each line represents a sample and each column a chemical compound. Each value in this table corresponded to the total ion count measured for a specific VOC in a given sample. Finally, to improve the comparison between VOC profiles, the probabilistic quotient normalization was applied [30] to suppress any effect related to different (overall) dilution of the samples.

### Multivariate data analysis

The data pre-processing resulted in a data table, reporting semi-quantitatively the concentration of 284 VOCs in 555 samples. Table 1 provides the complete description of all samples allocated to the three main factors influencing their VOC profiles: the four media used to culture the bacteria, the three types of bacteria and the resistance or sensitivity to antibiotic treatments. Some samples were lost due to incorrect sampling or analysis in those groups containing less than 24 samples.

The influence of these three factors was statistically evaluated using Multivariate Analysis of Variance (MANOVA). This analysis was performed using the implementation previously described [31], allowing to more efficiently deal with high number of variables (i.e. high number of detected VOCs). Significance was evaluated using Wilks' Lambda statistic. As the distribution of Wilks' Lambda under the null hypothesis was unknown here, a (sequential) permutation test was used to estimate this distribution from the data [32, 33].

### Compound identification

Selected VOCs were putatively chemically identified based on the NIST library and the analysis of the retention time and the mass spectrum by an experienced mass-spectrometrist.

### Results

In total 555 headspace samples were analyzed by GC-*tof*-MS. Data pre-processing of all chromatograms resulted in a data-matrix containing 284 VOCs.

Median chromatograms for all strains on each specific medium demonstrate clear differences in the VOC pattern excreted in the headspace by the antibiotic-resistant and–sensitive strains of the 3 selected bacteria cultured on the 4 different media. Fig 1 shows the median chromatograms of the antibiotic-sensitive strains on BHI as an example. All three microorganisms excreted a divergent peak pattern clearly indicating the different volatiles present in the headspace air of *S. aureus*, *P. aeruginosa* and *K. pneumonia* cultures.

**Table 1. Experimental design and sample distribution.**

| Medium | Bacteria | Antibiotic resistance | Number of samples |
|---|---|---|---|
| Brain heart infusion | _K. Pneumonia_ | + | 24 |
| | | - | 24 |
| | _P. Aeruginosa_ | + | 24 |
| | | - | 23 |
| | _S. Aureus_ | + | 17 |
| | | - | 18 |
| Marine broth | _K. Pneumonia_ | + | 24 |
| | | - | 24 |
| | _P. Aeruginosa_ | + | 24 |
| | | - | 24 |
| | _S. Aureus_ | + | 24 |
| | | - | 24 |
| Muller Hinton | _K. Pneumonia_ | + | 24 |
| | | - | 24 |
| | _P. Aeruginosa_ | + | 24 |
| | | - | 24 |
| | _S. Aureus_ | + | 24 |
| | | - | 24 |
| Tryptase Soy Agar | _K. Pneumonia_ | + | 24 |
| | | - | 24 |
| | _P. Aeruginosa_ | + | 24 |
| | | - | 22 |
| | _S. Aureus_ | + | 20 |
| | | - | 23 |

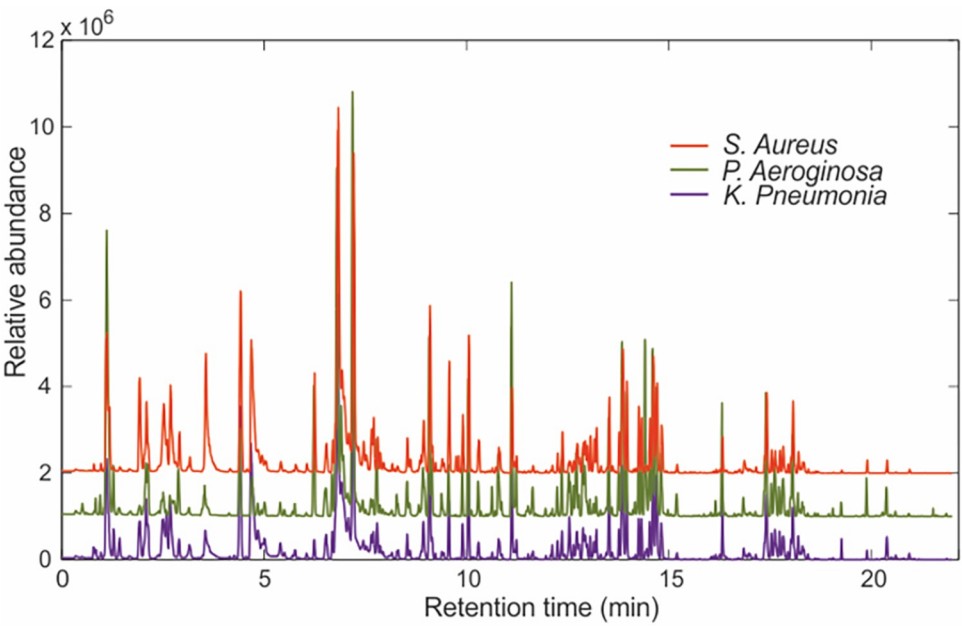

**Fig 1. Median chromatograms obtained with GC-tof-MS for the three antibiotic-sensitive bacterial strains cultured on BHI.**

**Table 2. Chemical identification of the 14 significantly different VOCs used to correctly classify the strains of the selected bacteria.** The reported change in abundance is with respect to the remaining groups.

| VOC | K Pneu | P Aero | S Aur |
| --- | --- | --- | --- |
| 2,4-dimethyl-1-heptene | ↓ | - | - |
| 2,4-dimethylpentane | - | ↓ | ↑ |
| 2-methylbutanal | - | ↓ | ↑ |
| Acetone | - | ↓ | ↑ |
| Acetophenone | ↑ | - | - |
| C9H20 | - | - | - |
| Decanal | ↓ | - | ↑ |
| Dimethyldisulfide | - | ↑ | ↓ |
| Ethanol | ↑ | - | - |
| Isobutanol | ↓ | ↑ | - |
| Isobutyric acid | - | ↓ | ↑ |
| Methanesulphonylchloride | ↓ | ↓ | ↑ |
| Pyrrole | ↑ | - | ↓ |
| Unknown | ↑ | - | - |

### Discriminating antibiotic-sensitive strains

MANOVA analysis using the obtained data matrix discriminated the 3 antibiotic-sensitive strains of *S. aureus*, *P. aeruginosa* and *K. pneumonia*, regardless of the used growth medium, based on the selection of 14 significantly different VOCs ($p<0.001$, Table 2). This separation is visualized in a score plot obtained using the canonical variables (linear combinations of the selected VOCs) of the MANOVA model (Fig 2A). The spread between the data points of the headspace samples of the bacteria indicates there is a reasonable variation within the excreted VOC profiles of each strain (Fig 2A). Most likely, this variation is introduced by the different growth media. However, this does not affect the distinction between the selected strains as can be observed in Fig 2B where recoloring the plot based on growth media did not result in clustering based on media type used.

The relationship of the 14 identified compounds towards the various microorganisms, and thus their contribution to an adequate identification of a sample, is shown in Table 2. Unlike all other compounds, only methanesulphonylchloride does not display a unique profile for only one of the strains. Instead, the abundance of methanesulphonylchloride is lower in the headspace samples of both *K. pneumonia* and *P. aeruginosa*, indicating that both bacteria excrete this volatile to a lesser extent or absorb it to a higher extent. All other significantly different VOCs show an abundance pattern unique for only one of the selected microorganisms, thereby strengthening the separation model visualized in Fig 2A and confirming the lack of influence of the various media depicted in Fig 2B.

To examine the descriptive nature of the selected VOCs with respect to the bacterial species in more detail, the influence of the significantly altered volatiles (as listed in Table 2) on the separation model is visualized in Fig 3. Within this plot, the lines indicate the changes in relative concentrations of the corresponding VOCs whereas the angle between compounds mirrors their correlation, i.e. a small angle means the compounds are positively correlated. For instance, the very small angle between ethanol and acetophenone indicates that the presence of these two volatiles will be influenced in the same way and that they will display similar trends for the three separated groups. Likewise, the very large angle between for instance decanal and pyrrole implicates that their concentrations will be affected in opposite ways in the samples of the different groups as can indeed also be deducted from Table 2. Additionally,

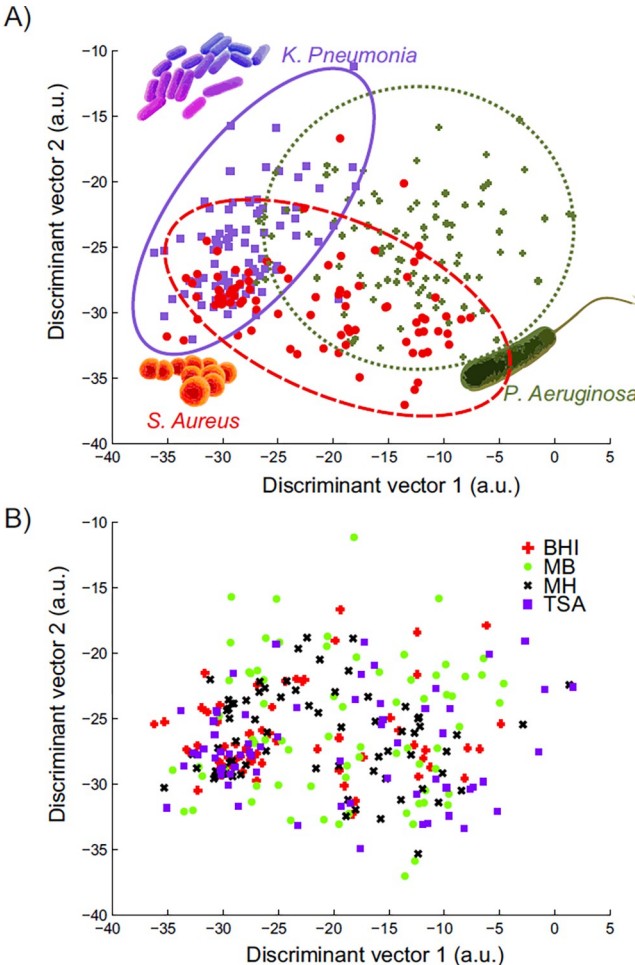

**Fig 2. Separation of the selected microorganisms on all types of media using VOC analysis.** A. Separation of the antibiotic-sensitive strains of S. aureus, *P. aeruginosa* and K. pneumonia based on 14 VOCs selected using MANOVA (p<0.001) and visualized with a score plot obtained using the canonical variables of this MANOVA model. B. Same visualization of all samples in the separation model but now recolored based on the applied media. *a.u. arbitrary unit.

the abundance of a compound is elevated if the sign of a new coordinate (i.e. discriminant vector) for samples and compound is the same (i.e. either both positive or both negative). As can be deducted from Fig 3, increased levels of especially decanal and 2-methylbutanal are indicative of the presence of *S. aureus* whereas elevated levels of mainly ethanol and acetophenone are associated with *K. pneumonia*. *P. aeruginosa* is defined by higher levels of particularly dimethylsulfide and isobutanol.

## Discriminating antibiotic-sensitive from -resistant strains

A significant discrimination between the antibiotic-sensitive and–resistant strains of *P. aeruginosa* is possible based on the complete VOC signature, consisting of 284 VOCs, of the headspace air samples of their cultures (Table 3). However, a distinction between the antibiotic-sensitive and–resistant strain of *S. aureus* and *K. pneumonia* could not be demonstrated. Although the different types of media resulted in significantly different VOCs excreted in the headspace air (p<0.001) of all three bacteria, no interaction of media was found for the discrimination their antibiotic sensitivity. This indicating rather subtle differences in VOC

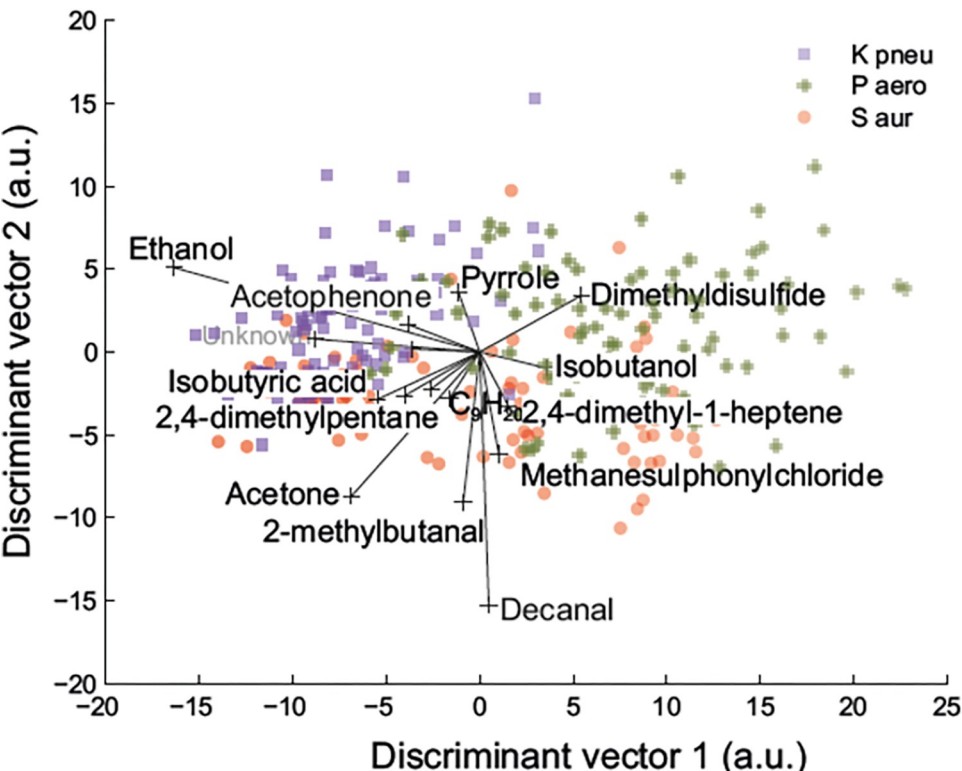

**Fig 3. Distinct VOCs determine the separation between the antibiotic-sensitive strains of S. aureus, *P. aeruginosa* and K. pneumonia.** *a.u. arbitrary unit.

profiles of antibiotic-sensitive and–resistant strains. However, for P-aeruginosa this interaction of the type of media does not influence the ability to discriminate between antibiotic–sensitive and -resistant strains, suggesting more obvious differences independent of the type of media used.

The separation model for the antibiotic-sensitive and–resistant strains of *P. aeruginosa* is visualized in a score plot using the canonical variables of the MANOVA model (Fig 4). Especially the antibiotic-sensitive strain displays a lot of variation in overall VOC signature excreted in the headspace air.

## Discussion

The results of this study demonstrate the potential of VOC analysis to identify the main causative bacteria for pneumonia *in vitro*. Indeed, all three selected microorganisms can be discriminated based on the relative abundance of unique VOCs in their headspace. Another important finding is that this discriminatory potential is not affected by the use of different growth

**Table 3. MANOVA outcome of the influence of antibiotic resistance on VOC signature of the selected bacteria.**

| p-values (MANOVA) | K. Pneumonia | S. Aureus | P. Aeruginosa |
|---|---|---|---|
| Antibiotic resistance | 0,993 | 0,16 | <0.001 |
| Media | <0.001 | <0.001 | <0.001 |
| Interaction | 1 | 0,32 | <0.001 |

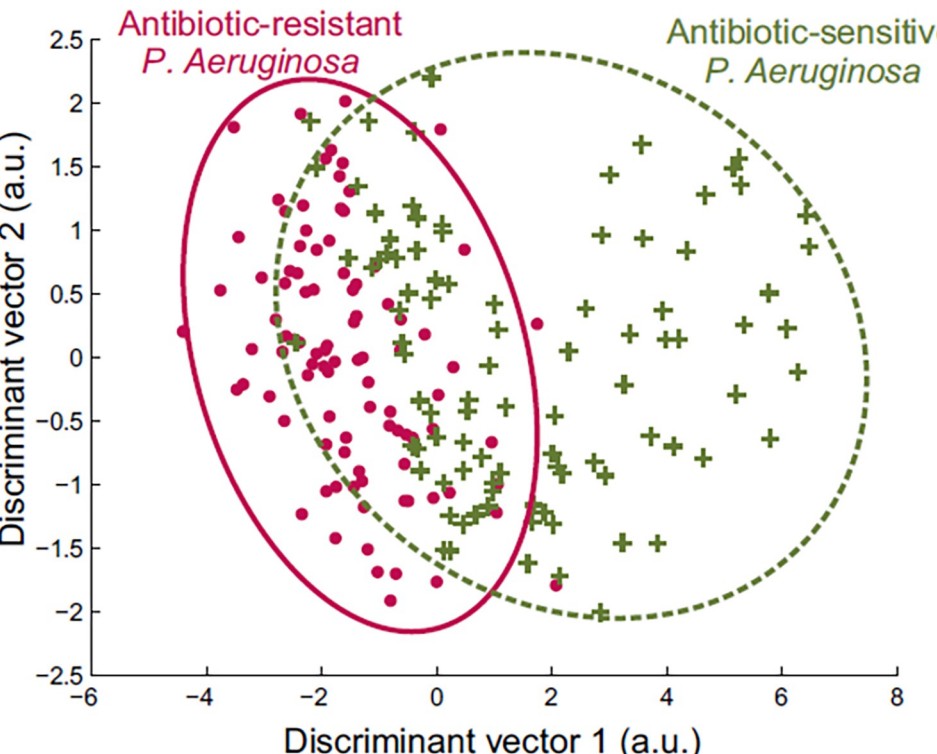

**Fig 4. Antibiotic resistance affects the unique VOC signature detected for *P. aeruginosa*.** Separation of the antibiotic-sensitive and–resistant strains of *P. aeruginosa* based on the overall VOC signature using MANOVA (p<0.001) and visualized with a score plot obtained using the canonical variables of this MANOVA model. *a.u. arbitrary unit.

media. Interestingly, VOC analysis is able to discriminate the antibiotic-resistant variant of *P. aeruginosa* from its antibiotic-sensitive variant. However, this distinction could not be observed for *S. aureus* and *K. pneumonia*.

Metabolic processes of *P. aeruginosa*, *S. aureus* and *K. pneumonia* result in the excretion of certain VOCs. Due to many different possible confounders on the production of VOCs it is possible that the VOCs identified in this study are less clear in other studies or that other VOCs are identified. VOCs in headspace air can be subject to different growth conditions such as the type of media used, the growth fase of the bacteria when collecting the headspace air. The current study indicates that 2-methylbutanal, a known aldehyde, to be produced by *S. aureus* specifically. This has been previously described in literature where both the isomers 2-methylbutanal and 3-methylbutanal are considered as distinctive compound for *S.aureus* [10, 12, 16, 34]. Additionally, the aldehyde decanal is also increased for *S.aureus* specifically. This specific increase of aldehydes can be a result of amino acid degradation by the bacteria [16].

Acetone, which was increased for *S.aureus* and decreased for *P.aeruginosa* in the current study, also appears to be increased *in vivo* in mice infected with *S.aureus* [35]. However, observed increases or decreases in abundance of compounds for individual bacteria are not always consistent and vary when conditions differ. For instance, in the current study, dimethyldisulfide is increased for *P. aeruginosa* and decreased for *S. aureus*, whereas this increase was only the case for *S. aureus* in our previous study [9]. A possible explanation for this apparent discrepancy is the dynamic process of bacterial growth resulting in changes in

the release of different compounds over time. A study, collecting headspace cultures on different time points, shows the presence of dimethyldisulfide earliest after 24h of bacterial growth for *P. aeruginosa* [16]. However, the growth time was similar in both studies as they cultured the bacteria overnight. An important different is that in the previous study only one type of growth media was used, while the current study used multiple. This could indicate that the type of growth media influences the VOCs produced. Despite the reported differences regarding its presence in the headspace air of *P. aeruginosa*, several studies have confirmed dimethyldisulfide can be considered a distinctive compound for this microorganism that potentially can be used in combination with other volatiles to identify its presence.

Besides endogenous pathways involved in the formation and excretion of microbial VOCs, exogenous factors such as growth media might also play a role in differences in VOCs produced by micro-organisms. It has been shown that nutrients present in the media can influence the growth pattern of the pathogens or lead to differences in VOCs produced [11, 34]. Some bacteria are able to grow in almost all kinds of growth media whereas others require very specific growth media. Different nutrients present may create a less or more optimal growth environment for the bacteria, which can subsequently be seen in a different growth curve. The nutrients in the growth media are metabolized by the bacteria resulting in different metabolic end-products and thus different volatiles excreted or absorbed. This indicates that compounds identified in studies using only one growth medium might not always be identified in other studies using different growth media. The current study, using 4 different growth media simultaneously, aimed to identify discriminative compounds that are independent of the growth media. For example, 3-hydroxy-2-butanone was identified previously in cultures of *S. aureus* using Trypticase Soy Broth but not in cultures using BHI or Nutrient Broth (NB) [11]. The fact that this compound was also not found in the current study further underlines the suggestion its presence depends on the growth medium used. Interestingly, dimethyldisulfide is previously identified for both *P.aeruginosa* and *S.aureus* in multiple growth media [16, 34] which also corresponds to the findings in the current study. Although the different types of growth media inevitably introduced variation within the excreted VOC profiles of the bacteria, the discriminatory value of the excreted VOCs in this study is still sufficient to enable discrimination between the respective strains. Moreover, the fact that strain-specific VOCs were identified despite the use of different growth media implicates that the discriminative volatiles are endogenously produced or absorbed and thus directly related to the micro-organism itself.

Both an antibiotic-sensitive and an antibiotic-resistant strain of the selected microorganisms were included in the current study. However, only for *P. aeruginosa* a separation could be made between the antibiotic-sensitive and–resistant type using all information available. Although our previous work did show a significant difference between antibiotic-resistant and–sensitive *S. aureus* grown on only Mueller-Hinton Broth [9], the current study was not able to repeat these results. The loss of samples due to incorrect sampling or analysis in especially the *S. aureus* group (n = 11 for antibiotic-resistant and n = 8 for the -sensitive subgroup), could possibly have lowered the statistical power for this analysis leading to a lack of statistical significance. Additionally, the use of multiple types of media created a larger overall variation within the groups which could have masked the differences associated with antibiotic resistance, indicating that those metabolic changes are subtle and thus more difficult to detect *in vitro*. An alternative approach to detect metabolic differences between antibiotic-sensitive and–resistant microorganisms might be to challenge their cultures with different classes of antibiotics as the genetic changes underlying antibiotic sensitivity might induce a more pronounced difference in response and thus in VOC excretion and absorption. Resistance towards methicillin, such as in *S. aureus*, is caused by the incorporation of a *mec* gene into the chromosome. This gene encodes for low affinity penicillin-binding proteins whose structure

causes limited binding capacity of methicillin resulting in resistance [36]. The resistant strains of *P. aeruginosa* and *K. pneumonia* are able to produce extended spectrum β-lactamase (ESBL) that is able to hydrolyze third-generation cephalosporins by altering the amino acid configuration around the active side of these β-lactamase resulting in resistance [37].

The bacteria applied in the present study differ in their structure as both *K. Pneumonia* and *P. aeruginosa* are so-called gram negative bacteria whereas *S. aureus* is a gram-positive microorganism. The cell envelope of gram-negative bacteria consists of a thin peptidoglycan wall that is surrounded by an outer lipid membrane containing lipopolysaccharide which is important in evading phagocytosis and cell lysis. Gram-positive bacteria have a thick peptidoglycan layer containing teichoic acids and no outer lipid layer. These teichoic acids of gram-positive bacteria may play a role in cell growth, preventing wall breakdown and possible cell lysis and provide much of the wall's antigenic specificity. As the cell membrane is involved in evading the actions of the immune system of a host and provides a barrier for certain antibiotics, differences in its structure might result in differences in response to environmental changes or to antibiotics [38]. The known differences between gram positive and gram negative bacteria most likely will affect which volatiles these microorganisms can absorb or excrete from specific growth media. However, the current study only studies the metabolic differences of bacterial growth in fixed culture media. These may be different in more complex *in vivo* situations in which multiple factors can influence the metabolic process which may result in more pronounced differences or the excretion of yet different VOCs. A study comparing both gram-negative and gram-positive bacteria, using metabolites in serum samples of infected mice, shows both a distinction between specific bacteria as well as overlapping characteristics between the different groups. For gram-positive bacteria metabolites such as, glucose, pyruvate, citrate, fumarate, and 2-oxoglutarate were identified, whereas for gram-negative bacteria higher levels of amino acids were found [35, 39].

Previous studies have already shown that VOCs produced in the headspace of *in vitro* pathogen cultures do not always completely correspond to the VOCs found in the exhaled breath of a diseased host [40]. Compared to standardized *in vitro* studies, living organisms have a more complex metabolism involving multiple factors such as diet and exercise [41] that confounds the possible interaction between host and pathogen. Although *in vitro* models cannot be extrapolated to *in vivo* situations, distinguishing various strains independent of their growth media is a step forward towards the clinical applicability of VOC analysis for quick identification of pathogens. Combining the results of multiple studies may provide insights in potentially overlapping, and thus more robust, VOCs. Additionally, dynamic *in* vitro sampling procedures can contribute to understanding metabolic changes during bacterial growth and can help identify bacteria-specific compounds independent of growth phases. Additionally, a next step could be the detection of pathogens in more complex models of metabolism, such as standardized animal studies or in clinical studies. As VOC analysis can be done on both *in vitro* (growth media or patient derived material) or *in vivo* (exhaled breath) samples, it could be a valuable tool for rapid and non-invasive detection of pathogens. Sampling devices for non-invasive breath collection in *in vivo* models facilitate in obtaining fundamental insight in the influence of metabolic processed on alterations in VOC composition and contribute to the translation of *in vitro* to *in vivo* results [42]. Such studies can contribute to an initial selection of important VOCs for the detection of specific pathogens *in vivo* allowing for more targeted analysis. Early diagnosis and identification of causative pathogens contributes to early administration of correct and specific antibiotics that would largely impact treatment, thereby increasing the recovery and survival of patients and decreasing their hospital stay and length of sickness.

## Supporting information

**S1 Data.**
(XLSX)

## Author Contributions

**Conceptualization:** Lionel Blanchet, Agnieszka Smolinska, Frederik-Jan van Schooten, Agnes W. Boots.

**Data curation:** Kim F. H. Hintzen, Marie-Louise Boumans, Ellen E. Stobberingh, Agnes W. Boots.

**Formal analysis:** Lionel Blanchet, Agnieszka Smolinska, Jan W. Dallinga.

**Funding acquisition:** Tim Lubbers, Frederik-Jan van Schooten.

**Investigation:** Kim F. H. Hintzen, Marie-Louise Boumans, Ellen E. Stobberingh, Jan W. Dallinga, Agnes W. Boots.

**Methodology:** Kim F. H. Hintzen, Agnieszka Smolinska, Jan W. Dallinga, Agnes W. Boots.

**Project administration:** Tim Lubbers, Frederik-Jan van Schooten, Agnes W. Boots.

**Resources:** Kim F. H. Hintzen, Marie-Louise Boumans, Ellen E. Stobberingh, Tim Lubbers, Frederik-Jan van Schooten.

**Software:** Agnes W. Boots.

**Supervision:** Tim Lubbers, Frederik-Jan van Schooten, Agnes W. Boots.

**Validation:** Lionel Blanchet, Agnieszka Smolinska, Jan W. Dallinga, Agnes W. Boots.

**Visualization:** Lionel Blanchet, Agnes W. Boots.

**Writing – original draft:** Kim F. H. Hintzen, Frederik-Jan van Schooten, Agnes W. Boots.

**Writing – review & editing:** Kim F. H. Hintzen, Agnieszka Smolinska, Tim Lubbers, Frederik-Jan van Schooten, Agnes W. Boots.

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
