## [Decision Letter · Decision Letter 0]

21 Jul 2023

PONE-D-23-12035Volatile Organic Compounds in headspace characterize isolated bacterial strains independent of growth medium or antibiotic sensitivityPLOS ONE

Dear Dr. Hintzen,

Thank you for submitting your manuscript to PLOS ONE. After careful consideration, we feel that it has merit but does not fully meet PLOS ONE’s publication criteria as it currently stands. Therefore, we invite you to submit a revised version of the manuscript that addresses the points raised during the review process.

We look forward to receiving your revised manuscript.

Kind regards,

Awatif Abid Al-Judaibi, PhD

Academic Editor

PLOS ONE

Journal Requirements:

a) The name of the colleague or the details of the professional service that edited your manuscript.

b) A copy of your manuscript showing your changes by either highlighting them or using track changes (uploaded as a *supporting information* file).

c) A clean copy of the edited manuscript (uploaded as the new *manuscript* file).

4. Please expand the acronym “MLDS” (as indicated in your financial disclosure) so that it states the name of your funders in full.

Reviewers' comments:

Reviewer's Responses to Questions

**Comments to the Author**

1. Is the manuscript technically sound, and do the data support the conclusions?

Reviewer #1: Yes

Reviewer #2: Yes

2. Has the statistical analysis been performed appropriately and rigorously? 

Reviewer #1: Yes

Reviewer #2: Yes

3. Have the authors made all data underlying the findings in their manuscript fully available?

Reviewer #1: Yes

Reviewer #2: No

4. Is the manuscript presented in an intelligible fashion and written in standard English?

Reviewer #1: Yes

Reviewer #2: Yes

5. Review Comments to the Author

Reviewer #1: The manuscripts addresses and important health issue and it is very relevant to human health. The authors have addressed the topic well and explained it in a way well understood to everyone. The tables and figures are well explained and clear.

Reviewer #2: I have reviewed the manuscript titled “Volatile Organic Compounds in headspace characterize isolated bacterial strains independent of growth medium or antibiotic sensitivity.”.

I recommend some revisions before acceptance and publication. Some of my comments are listed below:

Title: Article 'the' should be before headspace.

Abstract: The author should avoid the use of i.e.

Keywords: Volatile Organic Compounds and VOCs shouldn't be used as keywords at the same time.

Introduction: Good, but the author should use more recent references (within the last five years).

Materials and Methods:

The source of the bacterial isolates was not disclosed by the author

The author should be specific about the antibiotics used for the phenotyping test

Phenotypic and genotypic studies of bacterial isolates were not referenced

For clarity, the author should divide the number of samples in Table. 1 into different columns

Results:

The result of the minimum inhibitory concentration (MIC) of the specific antibiotics were not presented

The results of the genotypic analysis were not presented

The exact definitions of "K Pneu," "P Aero," and "S Aur" should be included in a key below Table 2.

Discussion: Tables and figures should be cited by the author when discussing

Conclusion: No conclusion was made by the author

References: More than half of the references should be recent (within the last five years)

6. PLOS authors have the option to publish the peer review history of their article (what does this mean?). If published, this will include your full peer review and any attached files.

Reviewer #1: **Yes: **Matrona Mbendo Akiso

Reviewer #2: **Yes: **AJAYI-ODOKO OMOLOLA

---

## [Author Response · Author response to Decision Letter 0]

29 Nov 2023

Reviewer #1

We thank the reviewer for the kind words and compliments.

Reviewer #2

Thank you for the extensive reviewing and the valuable comments made to the manuscript. We have carefully considered the comments and made the adjustments as described below.

1. Title: Article ‘the’ should be before headspace

Adjusted and replaced “the”

2. Abstract: The author should avoid the use of i.e. 

Deleted “i.e.”

3. Keywords: Volatile Organic Compounds and VOCs shouldn’t be used as keywords at the same time.

Replaced the individual keywords “volatile organic compounds” and “VOCs” by “volatile organic compounds (VOCs)”

4. Introduction: Good, but the author should use more recent references (within the last five years). 

We understand the importance of using recent references in order to provide up-to-date information. However, findings of studies older than 5 years also make a significant contribution to the description of results already achieved and the realization of this study. Within this relatively small field of research it is difficult to match the proposed ratio of references (more than half within the past 5 years). However, we have again searched the literature and found 9 published papers within the last five years and complemented the introduction. 

PAGE 3, LINE 10 

The most common causes of pneumonia are respiratory microorganisms including the bacteria Pseudomonas aeruginosa, Staphylococcus aureus and Klebsiella pneumonia.

REMOVED:

Parkins MD, Floto RA. Emerging bacterial pathogens and changing concepts of bacterial pathogenesis in cystic fibrosis. J Cyst Fibros. 2015;14(3):293-304.

NEW REFERENCE: 

Niederman MS, Torres A. Severe community-acquired pneumonia. Eur Respir Rev. 2022;31(166).

PAGE 3, LINE 13

New reference:

Torres A, Niederman MS, Chastre J, Ewig S, Fernandez-Vandellos P, Hanberger H, et al. International ERS/ESICM/ESCMID/ALAT guidelines for the management of hospital-acquired pneumonia and ventilator-associated pneumonia: Guidelines for the management of hospital-acquired pneumonia (HAP)/ventilator-associated pneumonia (VAP) of the European Respiratory Society (ERS), European Society of Intensive Care Medicine (ESICM), European Society of Clinical Microbiology and Infectious Diseases (ESCMID) and Asociación Latinoamericana del Tórax (ALAT). Eur Respir J. 2017;50(3).

PAGE 3, LINE 21 

Added text: 

Few in vitro studies already show the potential of VOCs analysis in headspace of bacterial cultures to differentiate between bacteria involved in pulmonary tract infections. 

New references:

Filipiak W, Żuchowska K, Marszałek M, Depka D, Bogiel T, Warmuzińska N, et al. GC-MS profiling of volatile metabolites produced by Klebsiella pneumoniae. Front Mol Biosci. 2022;9:1019290.

Kunze-Szikszay N, Euler M, Kuhns M, Thieß M, Groß U, Quintel M, et al. Headspace analyses using multi-capillary column-ion mobility spectrometry allow rapid pathogen differentiation in hospital-acquired pneumonia relevant bacteria. BMC Microbiol. 2021;21(1):69.

Steppert I, Schönfelder J, Schultz C, Kuhlmeier D. Rapid in vitro differentiation of bacteria by ion mobility spectrometry. Appl Microbiol Biotechnol. 2021;105(10):4297-307.

PAGE 3, LINE 29

Removed:

Robroeks CM, van Berkel JJ, Jöbsis Q, van Schooten FJ, Dallinga JW, Wouters EF, et al. Exhaled volatile organic compounds predict exacerbations of childhood asthma in a 1-year prospective study. Eur Respir J. 2013;42(1):98-106.

New references:

Ibrahim W, Natarajan S, Wilde M, Cordell R, Monks PS, Greening N, et al. A systematic review of the diagnostic accuracy of volatile organic compounds in airway diseases and their relation to markers of type-2 inflammation. ERJ Open Res. 2021;7(3).

Ratiu IA, Ligor T, Bocos-Bintintan V, Mayhew CA, Buszewski B. Volatile Organic Compounds in Exhaled Breath as Fingerprints of Lung Cancer, Asthma and COPD. J Clin Med. 2020;10(1).

van Vliet D, Smolinska A, Jöbsis Q, Rosias P, Muris J, Dallinga J, et al. Can exhaled volatile organic compounds predict asthma exacerbations in children? Journal of Breath Research. 2017;11(1):016016.

PAGE 3, LINE 33 

Removed:

Fowler SJ, Basanta-Sanchez M, Xu Y, Goodacre R, Dark PM. Surveillance for lower airway pathogens in mechanically ventilated patients by metabolomic analysis of exhaled breath: a case-control study. Thorax. 2015;70(4):320-5.

New reference:

Felton TW, Ahmed W, White IR, van Oort P, Rattray NJW, Docherty C, et al. Analysis of exhaled breath to identify critically ill patients with ventilator-associated pneumonia. Anaesthesia. 2023;78(6):712-21.

5. Materials and methods: 

- The source of the bacterial isolates was not disclosed by the author.

- The author should be specific about the antibiotics used for phenotypic test.

The source of the isolates is described in the methods on page 9 from line 8 onwards. These are the same ATCC reference strains and clinical isolates that were also used in our previous study and have added the reference on line 10.

- Phenotypic and genotypic studies of bacterial isolates were not referenced.

Indeed no references were provided for the phenotypic and genotypic studies. We have therefore added the following references in the methods on page 4:

Line 18

Rijnders MI, Deurenberg RH, Boumans ML, Hoogkamp-Korstanje JA, Beisser PS, Stobberingh EE. Population structure of Staphylococcus aureus strains isolated from intensive care unit patients in the netherlands over an 11-year period (1996 to 2006). J Clin Microbiol. 2009;47(12):4090-5.

Line 21

Stobberingh EE, Arends J, Hoogkamp-Korstanje JA, Goessens WH, Visser MR, Buiting AG, et al. Occurrence of extended-spectrum betalactamases (ESBL) in Dutch hospitals. Infection. 1999;27(6):348-54.

Line 22

Brink AA, von Wintersdorff CJ, van der Donk CF, Peeters AM, Beisser PS, Stobberingh EE, et al. Development and validation of a single-tube multiple-locus variable number tandem repeat analysis for Klebsiella pneumoniae. PLoS One. 2014;9(3):e91209.

- For clarity, the author should divide the number of samples in Table 1 into different columns. 

The table already distinguishes between the different bacteria and whether or not antibiotic resistance is present and provides the number of samples per separate group. In our opinion, adding additional columns will not contribute to a further clarification. We therefore propose to maintain the table in its current form. 

6 . Results:

- The result of the minimum inhibitory concentration (MIC) of the specific antibiotics were not presented 

This comment corresponds with comment number 5 requesting the source of the isolates and the antibiotics used. However, in order to further clarify this we have made the following adjustments in the methods: 

Removed: 

Antibiotic resistance was tested phenotypically by determining the minimum inhibitory concentration (MIC) of the specific antibiotics, e.g. methicillin and oxacillin for MRSA and MSSA and ESBL for the resistant and sensitive strains of both K. pneumonia and P. aeruginosa.

New text: 

Antibiotic resistance was tested phenotypically by determining the minimum inhibitory concentration (MIC) of the specific antibiotics. The MRSA and MSSA strains were provided from a previous study. For ESBL cefotaxim and ceftazidim were used with a cut-off value of > 1mg/L followed by a double disk diffusion with clavulanic acid to confirm ESBL production.

- The results of the genotypic analysis were not presented 

As discussed at comment 5, we have added references in the methods to further clarify the genotypic analysis. We therefore believe no further adjustments in the results are needed. 

- The exact definitions of “K Pneu,” “P Aero,” and “S Aur” should be included in a key below Table 2. 

We agree that for further clarification the exact definitions should be explained and have included an abbreviations section below Table 2. 

7. Discussion: Tables and figures should be cited by the author when discussing.

Indeed, it enhances the clarification and discussion of results when referred to the respective tables and figures. We have adjusted this in the discussion. 

8. Conclusion: No conclusion was made by the author.

“VOC analysis in the headspace of pathogens is a promising as rapid diagnostic tool for differentiating isolated micro-organisms independent of the applied growth media. Additionally, antibiotic-sensitive and -resistant strains of Pseudomonas aeruginosa, could be discriminated. However, validation of our results in appropriate in vivo models is critical to improve translation of breath analysis to clinical applications.”

9. References: More than half of the references should be recent (within the last five years).

As already discussed at “4. Introduction”, we understand the importance of using recent references in order to provide up-to-date information. A lot of research in the field of VOCs of pathogens has been published more than 5 years ago and we believe these studies are as important to mention as well as the research that is more recent. However, we have searched the literature and already adjusted the introduction as described above. Additionally we have made the following adjustments in the discussion: 

PAGE 14, LINE 17

Although the different types of growth media inevitably introduced variation within the excreted VOC profiles of the bacteria, the discriminatory value of the excreted VOCs in this study is still sufficient to enable discrimination between the respective strains. Moreover, the fact that strain-specific VOCs were identified despite the use of different growth media implicates that the discriminative volatiles are directly related to the micro-organism itself. 

Added text:

These findings are consistent with other studies in which multiple types of growth media were evaluated for the discrimination of bacteria based on VOCs in the headspace of S. aureus and P. aeruginosa. 

New references: 

Fitzgerald S, Holland L, Morrin A. An Investigation of Stability and Species and Strain-Level Specificity in Bacterial Volatilomes. Front Microbiol. 2021;12:693075.

Jenkins CL, Bean HD. Influence of media on the differentiation of Staphylococcus spp. by volatile compounds. Journal of Breath Research. 2019;14(1):016007.

PAGE 16, LINE 3

Although in vitro models cannot be extrapolated to in vivo situations, distinguishing various strains independent of their growth media is a step forward towards the clinical applicability of VOC analysis for quick identification of pathogens.

Added text: 

An alternative step in improving the translation of the results from in vitro to in vivo could be the headspace analysis of bacteria in whole blood cultures as a more representative growth method to in vivo situations. One study showed that distinct VOCs could already be identified after 6 hours of incubating E coli, S aureus or P aeruginosa in whole blood samples. Another study detected bacteria specific VOCs within 8 hours of culturing whole blood samples from animals with an E coli induced sepsis. However, this only applies to more widespread infections involving bacteremia and may therefore be less suitable for early detection of infections. 

New references: 

Drees C, Vautz W, Liedtke S, Rosin C, Althoff K, Lippmann M, et al. GC-IMS headspace analyses allow early recognition of bacterial growth and rapid pathogen differentiation in standard blood cultures. Appl Microbiol Biotechnol. 2019;103(21-22):9091-101.

Euler M, Perl T, Eickel I, Dudakova A, Maguilla Rosado E, Drees C, et al. Blood Culture Headspace Gas Analysis Enables Early Detection of Escherichia coli Bacteremia in an Animal Model of Sepsis. Antibiotics (Basel). 2022;11(8).

---

## [Editor Report · Decision Letter 1]

27 Dec 2023

Volatile Organic Compounds in headspace characterize isolated bacterial strains independent of growth medium or antibiotic sensitivity

PONE-D-23-12035R1

Dear Dr. Kim F.H. Hintzen,

We’re pleased to inform you that your manuscript has been judged scientifically suitable for publication and will be formally accepted for publication once it meets all outstanding technical requirements.

Kind regards,

Awatif Abid Al-Judaibi, PhD

Academic Editor

PLOS ONE

---

## [Editor Report · Acceptance letter]

15 Jan 2024

PONE-D-23-12035R1 

PLOS ONE

Dear Dr. Hintzen, 

I'm pleased to inform you that your manuscript has been deemed suitable for publication in PLOS ONE. Congratulations! Your manuscript is now being handed over to our production team.

Kind regards, 

on behalf of

Professor Awatif Abid Al-Judaibi 

Academic Editor

PLOS ONE